# Development of Machine Learning Models Predicting Estimated Blood Loss during Liver Transplant Surgery

**DOI:** 10.3390/jpm12071028

**Published:** 2022-06-23

**Authors:** Sujung Park, Kyemyung Park, Jae Geun Lee, Tae Yang Choi, Sungtaik Heo, Bon-Nyeo Koo, Dongwoo Chae

**Affiliations:** 1Department of Anesthesiology and Pain Medicine, Anesthesia and Pain Research Institute, Yonsei University College of Medicine, 50-1 Yonsei-ro, Seodaemun-gu, Seoul 03722, Korea; mdpark@yuhs.ac; 2Center for Viral Immunology, Korea Virus Research Institute, Institute for Basic Science, 55 Expo-ro, Doryong-dong, Yuseong-gu, Daejeon 34126, Korea; pkm304@gmail.com; 3Department of Transplantation Surgery, Yonsei University College of Medicine, 50-1 Yonsei-ro, Seodaemun-gu, Seoul 03722, Korea; drjg1@yuhs.ac; 4Department of Anesthesiology and Pain Medicine, National Health Insurance Service Ilsan Hospital, Goyang-si 10444, Gyeonggi-do, Korea; redsunkak@nhimc.or.kr (T.Y.C.); zestever93@yuhs.ac (S.H.); 5Department of Pharmacology, Yonsei University College of Medicine, 50-1 Yonsei-ro, Seodaemun-gu, Seoul 03722, Korea

**Keywords:** estimated blood loss, liver transplantation, machine learning

## Abstract

The incidence of major hemorrhage and transfusion during liver transplantation has decreased significantly over the past decade, but major bleeding remains a common expectation. Massive intraoperative hemorrhage during liver transplantation can lead to mortality or reoperation. This study aimed to develop machine learning models for the prediction of massive hemorrhage and a scoring system which is applicable to new patients. Data were retrospectively collected from patients aged >18 years who had undergone liver transplantation. These data included emergency information, donor information, demographic data, preoperative laboratory data, the etiology of hepatic failure, the Model for End-stage Liver Disease (MELD) score, surgical history, antiplatelet therapy, continuous renal replacement therapy (CRRT), the preoperative dose of vasopressor, and the estimated blood loss (EBL) during surgery. The logistic regression model was one of the best-performing machine learning models. The most important factors for the prediction of massive hemorrhage were the disease etiology, activated partial thromboplastin time (aPTT), operation duration, body temperature, MELD score, mean arterial pressure, serum creatinine, and pulse pressure. The risk-scoring system was developed using the odds ratios of these factors from the logistic model. The risk-scoring system showed good prediction performance and calibration (AUROC: 0.775, AUPR: 0.753).

## 1. Introduction

Although liver transplantation (LT) has emerged as the treatment of choice for patients with end-stage liver disease, patients who undergo LT experience a variety of complications, including infection, mortality, and surgical re-intervention [1,2]. The incidence of major hemorrhage and transfusion during LT over the past decade has decreased significantly [3], but major bleeding during surgery is commonly expected [4,5]. Post-transplantation complications are associated with intraoperative massive hemorrhage.

The classical definition of massive hemorrhage is the loss of one blood volume within a 24-h period. Correspondingly, massive transfusion in an adult has commonly been defined as the use of 10 or more units of packed red blood cells in a 24-h period, which approximates the replacement of one blood volume, i.e., the blood volume for a 70-kg male [6,7] of approximately 5 L [8]. The Joint United Kingdom Blood Transfusion and Tissue Transplantation Services Professional Advisory Committee also defines major hemorrhage as either the loss of over one blood volume (70 mL/kg, or more than 5 L in a 70 kg adult) in 24 h, the loss of 50% of the total blood volume in less than 3 h, or bleeding at a rate greater than 150 mL/min [9]. Therefore, more than 5 L of blood loss indicates massive hemorrhage that requires massive transfusion.

Risk assessment of massive bleeding before LT can guide surgical preparation and patient management. For example, large-bore c-line catheterization and continuous blood pressure monitoring are required before surgery [10,11]. If the amount of bleeding can be predicted preoperatively, unnecessary transfusion preparation or catheterization can be reduced [12]. In addition, targeted preventive measures can be implemented for high-risk patients. Although multiple risk factors for massive bleeding requiring transfusion during LT exist, few predictive models have been constructed incorporating these risk factors.

Machine learning (ML) approaches have been found useful in medical science for the prediction of clinical outcomes from complex multifactorial mechanisms [13]. Moreover, ML can efficiently assess the relevance of a large number of variables to the outcome of interest, and can thus build predictive models. Such models can be deployed in real-world situations, including clinical decision making.

This study aims to use ML to develop a prediction model and risk-scoring system for massive hemorrhage during LT. We believe that this system could help to identify high-risk patients and develop preventive strategies to reduce the occurrence of massive bleeding in LT patients.

## 2. Methods

### 2.1. Patients and Data Collection

Data were collected from patients aged >18 years who underwent LT between January 2015 and January 2020 at Severance Hospital. The Severance Hospital Institutional Review Board (IRB #4-2020-1157) granted consent waivers based on the general impracticability and minimal harm.

Currently, the common operative methods for LT include classical LT, piggyback LT, and classical venous bypass LT. Most patients with living donors in our study underwent piggyback LT. The major advantage of this method is reduced bleeding. The patients with deceased donors underwent classical LT.

The physiological parameters induced by end-stage liver disease were collected via electrical medical records (EMR) (Appendix A). Preoperative variables, such as the dose and quantity of vasopressors and vital signs, were collected within 24 h of the day of surgery. For variables with multiple measurements, the values at the ward or intensive care unit closest to the surgery start time were chosen. A single measurement of the last vital signs before departing to the operating room in the ward or intensive care unit was chosen. The other parameters used were emergency surgery; information on the donor (weight, robot surgery, living or cadaver); demographic data (age, sex, height, weight); the etiology of hepatic failure (viral hepatitis B or C, hepatocellular carcinoma, alcoholic liver disease, liver cirrhosis); the Model for End-stage Liver Disease (MELD) score; surgical history; the incidence of antiplatelet therapy, continuous renal replacement therapy (CRRT), and preoperative radiation therapy; thrombosis or collateral on a CT scan; preoperative transfusion; the antibody titer at the preoperative test; and planned splenectomy. The estimation of blood loss was recorded every hour during LT from the amount of irrigation, the fluid collected in the suction bottle, and the count of wet gauze used. The total blood loss was automatically collated and recorded in the EMR. Cases in which LT was received multiple times in a single patient or canceled due to unstable hemodynamic signs during surgery were excluded.

### 2.2. Statistical and Machine Learning Methods

#### 2.2.1. Data Preprocessing

The data were split into a training set (70%) and a test set (30%). The feature selection and model training were conducted on the training set, while the test dataset was used to assess the predictive performance of the trained models. Hemoglobin (Hb) was substituted for hematocrit (Hct). Because systolic blood pressure (SBP) and diastolic blood pressure (DBP) were highly correlated, mean arterial pressure (MAP) and pulse pressure (SBP and DBP) were introduced as markers of large arterial rigidity [14,15,16]. The outcome, estimated blood loss (EBL), was categorized as massive bleeding for EBL ≥ 5000 cc and no massive bleeding for EBL < 5000 cc.

#### 2.2.2. Feature Selection

The features were selected using stability selection, a LASSO-based method implemented in the R *stabs* (version 0.6-4) package in nested cross-validation (Figure 1) [17]. In brief, this method assesses the stability scores for each feature, quantified as the frequency of inclusion among the bootstrapped subsamples. This metric indicates the robustness of each feature selection from the given random sampling variations. For each outer training fold, the features were ranked in order of decreasing stability score. The logistic regression models were then evaluated from the fivefold inner cross-validated predictive performance by incrementally adding top-ranking features. After all of the iterations across the outer folds, consensus stability scores were obtained for all of the features by pooling all of the stability scores. The optimal number of features was then determined as the minimum number of features associated with the maximum averaged cross-validated predictive performance. Then, the features with consensus stability scores up to the optimal number rank were selected. These features were then further filtered using multivariable logistic regression to select only the significant features (*p*-value < 0.1).

#### 2.2.3. Machine Learning Model Training

Based on the selected features, we compared the predictive performance of six machine learning models trained using logistic regression, elastic net, support vector machines (SVM), random forests (RF), extreme gradient boosting (Xgboost), and neural networks (NN) via the fivefold cross-validation of the training dataset. The predictions were assessed using the areas under the receiver characteristic curve (AUROC) and the precision–recall curve (AUPR) based on the test dataset. The R *caret* (version 6.0-91), *xgboost* (version 1.5.2.1), and *Keras* (version 2.8.0) packages were used.

#### 2.2.4. Risk Scoring System Development

The risk scoring system was developed by first discretizing the continuous features into categorical ones, then using the category regression coefficients to assign appropriate risk scores. In order to find the optimal categorizing boundaries, we first obtained the deciles of each of the continuous features, generating nine binary categorization schemes, then retrained fivefold cross-validated logistic regression models for each categorization scheme. We then selected the scheme with the highest cross-validated AUROC. The same procedure was applied recursively in order to determine additional bins for each of the continuous features, not until the cross-validated AUROC improved. The categorized features were subsequently filtered based on the statistical significance of the regression coefficients and their effects on the Akaike information criteria. Finally, the feature scores were derived by finding the corresponding integer values that best preserved the relative ratios among the regression coefficients.

## 3. Results

Initially, 418 patients were included in this study. After excluding four patients with cardiac arrest or malignancy whose LTs were canceled, and whose EBL was extremely high, 414 patients remained. We pre-processed the collected data by binarizing EBL into massive bleeding (EBL ≥ 5000 cc) or no massive bleeding (EBL < 5000 cc) (Table 1). The correlation heatmap revealed several clusters of correlated features, one of which included the EBL (Appendix A). This suggests that input features can inform EBL but may suffer collinearity among features when building predictive models.

The most predictive features were selected next (Figure 1). We utilized LASSO-based stability selection for nested cross-validation in order to address collinearity among features and avoid false-positive feature selection. Initially, the features were ranked according to consensus stability scores (Figure 2A). The optimal rank cut-off of 10 was determined as the number of features that generated the highest averaged cross-validation AUROC across the outer training fold (Figure 2B). After additional filtering through multivariable logistic regression analysis, eight features were finally selected: hepatocellular carcinoma (HCC) status, aPTT, operation time, body temperature, MELD, MAP, creatinine, and pulse pressure.

Machine learning models were then built using these selected features. We compared the prediction performances for the test dataset of models trained using different machine learning methods (Table 2, Appendix A). Multivariable logistic regression models, neural networks, and an SVM with a linear kernel had the best performances of the models tested. Therefore, we chose the multivariate logistic model as the most interpretable of the three high-performance options (Table 3).

We developed risk-scoring systems to predict the probability of massive bleeding in order to maximize the clinical utility of the final logistic regression model. Firstly, an optimally categorized logistic regression model was derived and its adequate prediction performance was confirmed (AUROC—0.776; AUPR—0.760). We then derived a scoring system from the feature coefficients of the categorized model (Table 4) and mapped the scores with the probability of massive bleeding (EBL ≥ 5000 cc) (Figure 3). The risk-scoring system showed good prediction performance and calibration (AUROC—0.775; AUPR—0.753; Appendix A), delineating a 17-fold difference in the probability of massive bleeding across the score intervals.

## 4. Discussion

This study aimed to predict massive bleeding during LT. The most important factors for the prediction of massive hemorrhage were the disease etiology, aPTT, operation duration, body temperature, MELD score, mean arterial pressure, serum creatinine, and pulse pressure.

Massive hemorrhage and transfusion during LT have been associated with higher mortality and prolonged hospital stays [18]. Perioperative bleeding and transfusion remain potent predictors of postoperative complications and graft survival [4]. Therefore, it is important to predict massive bleeding and prepare for transfusion before surgery.

The MELD was initially created in order to predict survival in patients with complications undergoing the elective placement of transjugular intrahepatic portosystemic shunts. MELD was associated with patients requiring blood products [19], but was a poor predictor of blood loss or blood transfusion in previous studies [20,21]. However, Yoon et al. [22] found that a MELD score of 10.5 was the cut-off value to distinguish the intraoperative transfusion and no transfusion groups. In our study, the patients had a higher probability of massive bleeding and transfusion for MELD scores > 10 points. This may be insufficient to predict bleeding only with the MELD score, but when considered together with other factors, the MELD score can be considered an important factor in massive hemorrhage prediction.

The MELD score is calculated based on the serum creatinine, bilirubin, and prothrombin time-international normalized ratio (PT-INR). According to the initiation and termination of the massive transfusion protocol [23], viscoelastic hemostatic assay-guided resuscitation is superior to conventional coagulation assay-guided resuscitation in hemorrhaging trauma patients. However, not all hospitals can use viscoelastic hemostatic assays in clinical settings due to economic and technical constraints. PT-INR and aPTT are more commonly used to assess coagulation profiles before surgery. Although the viscoelastic assay is suitable for the intraoperative period, our risk scoring system using laboratory values such as aPTT and MELD is both practical and intuitive for the preoperative prediction of massive hemorrhage.

Many studies [24,25,26] have shown that a high PT-INR does not result in large amounts of bleeding. For example, PT-INR values were not correlated with the bleeding risk in cirrhotic patients [27]. Our study also found that PT-INR was inappropriate for the determination of the risk of massive bleeding during surgery.

Previous studies have reported that aPTT was predictive of major bleeding in cirrhotic patients [28,29]. Our results were consistent with those of the previous reports. However, some papers have shown that coagulation defects of PT-INR or aPTT are inadequate to reflect the balance of coagulation due to the rebalancing of coagulation with increased von Willebrand factor and factor VIII, and decreased natural anticoagulant proteins in patients with end-stage liver disease [30,31,32,33]. Further investigation is necessary in order to predict the bleeding risk and evaluate the coagulation status in end-stage liver disease.

Hypothermia progressively impairs platelet aggregability and clot formation [34]. Therefore, it was confirmed that hypothermia-induced coagulopathy led to massive hemorrhage. The maintenance of intraoperative normothermia reduces blood loss and allogeneic blood requirements [12]. As preoperative hypothermia will continue during the intraoperative period, it may be a risk factor for massive bleeding.

Pulse pressure is defined as the difference between diastolic and systolic blood pressures. A narrow pulse pressure indicates an early response to decreased intravascular volume [35], and increases the predicted probability of active hemorrhage. In trauma patients, a narrow pulse pressure is independently associated with a threefold increase in significant transfusions [36,37]. As normal pulse pressure is 30–40 mmHg, the cut-off value of 55 mmHg suggested in our study is higher; however, the intravascular volume is decreased for values below this cut-off.

End-stage liver disease leads to peripheral vasodilation due to increased NO production and cardiac output. MAP is reduced as myocardial contractility decreases due to cirrhotic cardiomyopathy resulting from this vasodilator-mediated hyperdynamic circulatory state. MAP may also decrease in patients with variceal bleeding before surgery. Our experimental results confirmed that massive bleeding and transfusion could be expected in patients with hypovolemia or preoperative hypotension conditions. If hypotension or hypovolemia is improved before surgery, massive bleeding and transfusion can be reduced, thereby improving patient outcomes.

A prolonged operative duration can result from a technical problem during surgery or from controlling a massive hemorrhage. Additionally, as the operation time increases, bleeding may continue (even in small amounts) at the cut surface of the transplanted liver. Previous episodes of variceal bleeding, previous abdominal surgery, and surgeon experience were associated with increased operating duration. These factors, excluding surgeon experience, were included in our analysis, but were not valuable in the prediction of blood loss for our study. Conversely, if the patient’s anterior–posterior diameter is large, or if collateral not visible on CT or MRI scans is found during the operation, the operation duration increases and massive hemorrhage occurs.

An increased bleeding tendency has been associated with various degrees of renal insufficiency. Renal insufficiency results from multifactorial causes but affects all of the aspects of platelet function, including adhesion, secretion, and aggregation [38]. According to Modanlou et al. [39], transplantation is almost four times more likely to require 10 units of RBC in cases in which the recipient patient has a pre-surgery serum creatinine level greater than 1.3 mg/dL. This is consistent with our conclusion that higher creatinine levels increase the hemorrhage risk during surgery.

In a study by Danforth et al. [18], patients with HCC had a high probability of massive bleeding. However, in our study, patients with HCC had a low probability of massive bleeding. This is because our pre-cirrhotic HCC patients had relatively good vascularity conditions and coagulation profiles. The average MELD score of HCC patients in our study was 12.3 points.

One limitation of this study is that it is a single-center retrospective review. However, the guidelines for transfusion and the estimation of blood loss are standardized across healthcare locations. Another limitation was our relatively small population size of 414 patients, with relatively many features included.

In conclusion, we have developed a machine learning model and scoring system to predict massive hemorrhage risk during LT. This model could be helpful for clinicians when deciding on anesthetic management choices for LT.

## Figures and Tables

**Figure 1 jpm-12-01028-f001:**
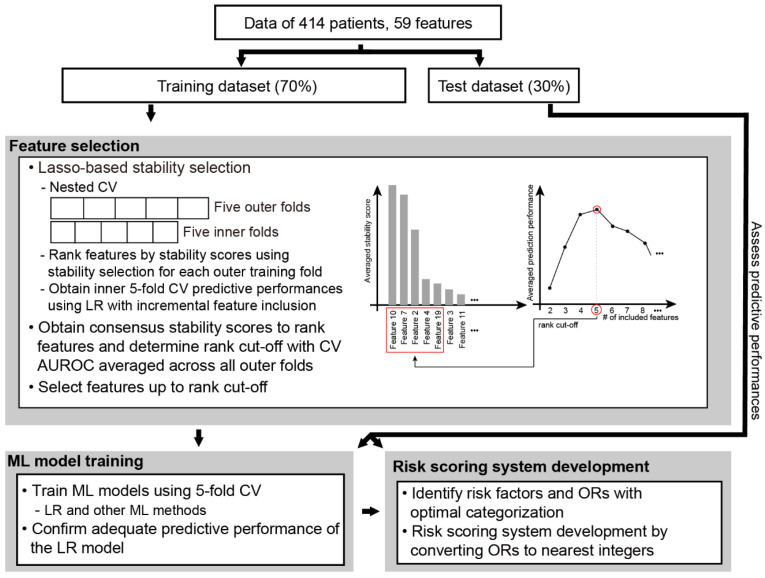
Overall workflow. CV, cross-validation; LR, logistic regression; OR, odds ratio.

**Figure 2 jpm-12-01028-f002:**
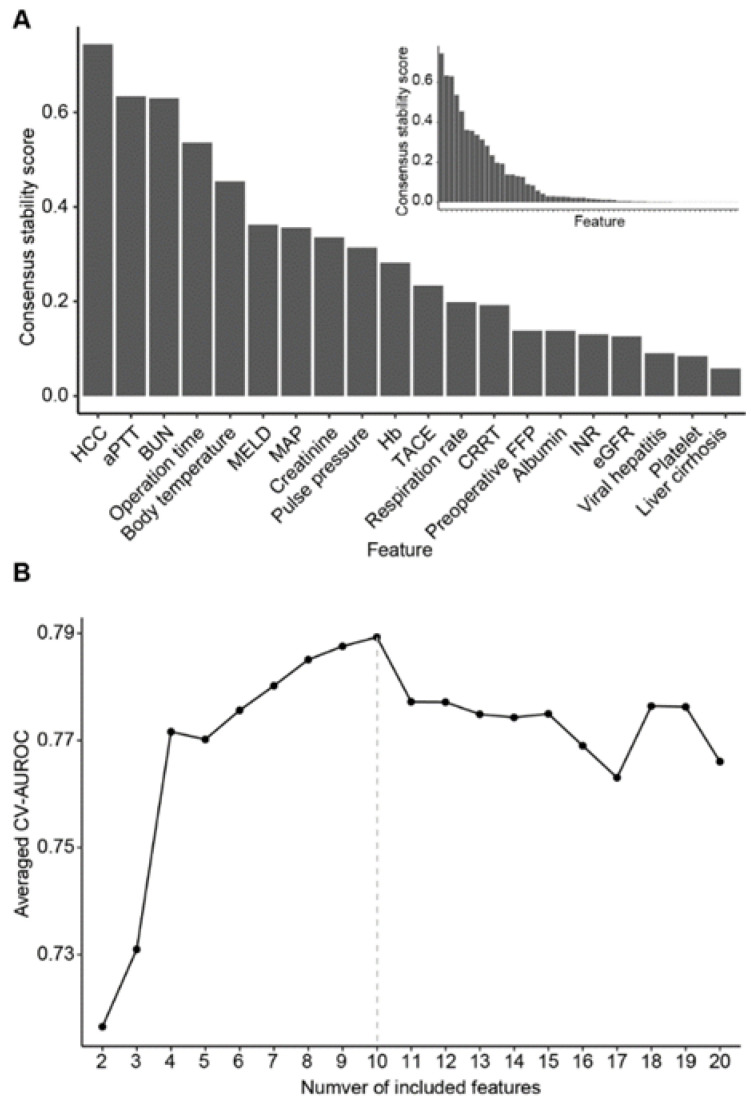
Feature selection results. (**A**) Features ordered according to consensus stability scores averaged across the outer CV folds (Methods). Outset: The top 20 features are shown. Selected features based on the optimal number of features are in bold. Inset: The overall distribution of the stability scores of all of the features is shown. (**B**) Averaged CV-AUROC with the incremental numbers of the features. The grey vertical line indicates the optimal number of features.

**Figure 3 jpm-12-01028-f003:**
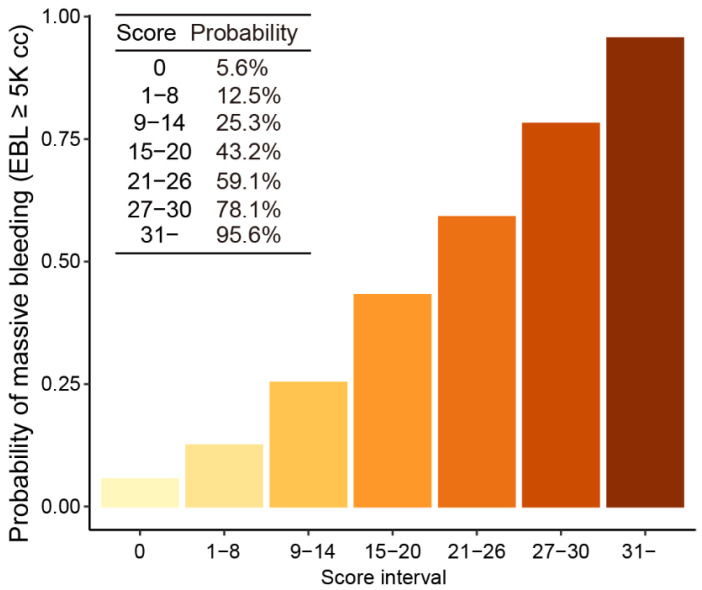
Probabilities of massive bleeding (EBL ≥ 5000 cc) for the score intervals. 5K, 5000.

**Table 1 jpm-12-01028-t001:** Patients’ demographics.

	EBL < 5000 cc (N = 228)	EBL ≥ 5000 cc (N = 186)	*p* Value
EBL (mean (SD))	2753.6 (1258.0)	8755.1 (4220.7)	<0.001
Age (mean (SD))	56.07 (8.5)	55.43 (9.1)	0.458
Sex = male (%)	159 (69.7)	132 (71.0)	0.869
Height (mean (SD))	165.20 (8.1)	165.27 (8.9)	0.932
Weight (mean (SD))	65.28 (11.0)	66.67 (11.8)	0.216
Emergency (%)	43 (18.9)	51 (27.4)	0.051
Cadaver donor (%)	41 (18.0)	44 (23.7)	0.194
Operation time (mean (SD))	617.5 (138.5)	666.2 (168.8)	<0.001
Liver cirrhosis (%)	141 (61.8)	148 (79.6)	<0.001
Alcoholic liver disease (%)	60 (26.3)	77 (41.4)	0.002
Hepatocellular carcinoma (%)	143 (62.7)	65 (34.9)	<0.001
MELD (mean (SD))	13.45 (8.5)	19.08 (10.7)	<0.001
Hb (mean (SD))	10.93 (2.2)	9.76 (2.0)	<0.001
Hct (mean (SD))	32.22 (6.3)	28.81 (5.8)	<0.001
Platelet (mean (SD))	94.60 (53.3)	80.98 (54.3)	0.011
PT (mean (SD))	1.4 (0.6)	1.7 (0.5)	<0.001
aPTT (mean (SD))	40.0 (13.2)	51.4 (30.0)	<0.001
Albumin (mean (SD))	3.1 (0.6)	2.9 (0.5)	<0.001
Blood Urea Nitrogen (mean (SD))	17.6 (11.0)	27.3 (23.0)	<0.001
Creatinine (mean (SD))	0.8 (0.4)	1.2 (1.1)	<0.001
eGFR (mean (SD))	87.0 (43.0)	56.2 (40.9)	<0.001
Mean arterial pressure (mean (SD))	91.1 (10.6)	86.5(10.7)	<0.001
Body temperature (mean (SD))	36.6 (0.4)	36.5 (0.4)	<0.001
Pulse pressure (mean (SD))	57.5 (12.5)	52.9 (12.6)	<0.001

**Table 2 jpm-12-01028-t002:** Prediction performances of the trained machine learning models.

Machine Learning Method	Test Dataset	
	AUROC	AUPR
Multivariable logistic regression	0.840	0.821
Elastic net	0.764	0.678
Random forests	0.803	0.783
Extreme gradient boosting	0.806	0.797
Neural networks	0.851	0.804
SVM with radial kernel	0.832	0.804
SVM with linear kernel	0.841	0.818

AUROC, area under the receiver operating characteristic curve; AUPR, area under the precision–recall curve.

**Table 3 jpm-12-01028-t003:** Final logistic regression model.

	Coefficient	Odd Ratio	95% CI	*p*-Value
HCC	−0.93	0.39	(0.22–0.69)	0.001
aPTT per 10 s	0.21	1.23	(1.05–1.49)	0.015
Operation time per hour	0.29	1.33	(1.17–1.52)	<0.001
Body temperature per 0.5 °C	−0.41	0.66	(0.46–0.95)	0.026
MELD per 10	0.17	1.19	(0.99–2.00)	0.055
MAP per 10 mmHg	−0.30	0.74	(0.56–0.97)	0.033
Creatinine per 0.5	0.36	1.44	(1.09–2.06)	0.027
Pulse pressure per 10 mmHg	−0.29	0.75	(0.59–0.94)	0.015

**Table 4 jpm-12-01028-t004:** Risk scoring system.

		Scores
HCC		
	No	+5
aPTT (sec)		
	≥40	+5
operation time (min)		
	≥630 and <810	+7
	≥810	+12
body temperature (°C)		
	<36.3	+9
MELD		
	≥10	+4
MAP (mmHg)		
	<70	+14
creatinine (mg/dL)		
	≥0.95 and <1.15	+8
	≥1.15	+10
pulse pressure (mmHg)		
	<55	+4

## Data Availability

The data presented in this study are available on request from the corresponding author. The data are not publicly available due to privacy.

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
