# Peer review of "Development of Machine Learning Models Predicting Estimated Blood Loss during Liver Transplant Surgery"

_jpm, 2022, doi:10.3390/jpm12071028_

Round 1

Reviewer 1 Report

In this manuscript, the authors applied a range of machine learning methods to predict massive blood loss in patients during liver transplantation surgery. They selected logistic regression as the best interpretable model, and further develop a risk-scoring system. The results showed good prediction performance and calibration (AUROC: 0.775, AUPR: 0.753) in predicting massive hemorrhage. However, there are some critical points for their conclusion that need to be addressed by the authors. Below are more specific comments by sections:

 1.      Introduction:

(1)    For literature review, more evidence regarding machine learning methods can be provided by adding relevant studies, such as

Liu, L. P., et al. (2021). Machine learning for the prediction of red blood cell transfusion in patients during or after liver transplantation surgery. Frontiers in medicine, 8, 81.

McCluskey SA, et al. (2006). Derivation of a risk index for the prediction of massive blood transfusion in liver transplantation. Liver Transpl 12:1584–93. doi: 10.1002/lt.20868.

Pustavoitau A, et al. (2017) Predictive modeling of massive transfusion requirements during liver transplantation and its potential to reduce utilization of blood bank resources. Anesth Analg. 124:1644–52. doi: 10.1213/ANE.0000000000001994

 Besides, this references can be discuss in Discussion.

 2.      Methods:

Section 2.1:

(1)    Please give the total number of candidate predictors. Is it 59 shown in Figure 1? The author can provide a list of all predictors in Appendix.

Section 2.2.2:

(1)    It appears that the feature selection method in favor of logistic regression model (e.g., These features were then further filtered using multivariable logistic regression.). With regard to the impact of feature selection might alter their findings, I suggest to include all significant predictors, 20 variables in Table 1, to re-run the prediction using these ML methods.  

 Section 2.2.3

(1)    The setting of parameters for each machine learning method (i.e., elastic net, RF..etc.) should be given.

(2)    Typo: linear regression? or logistic regression

Section 2.2.4:

(1)    It is difficult to understand the scoring system. Please provide a diagram for this.

(2)    Categorizing continuous data looks pretty similar with the rationale of Tree-based methods, e.g., RF or Xgboost. The authors should compare the risk scoring system with that obtained from these tree-based methods.

(3)    The authors can discuss the impact of small sample size (e.g., 30 % of validation data) on risk scoring.

 3.      Results:

(1)    In Table 2. Logistic regression with AUROC=0.84 and AUPR=0.821. The final logistic model is presented in Table 3. It is not clear that “Firstly, an optimally categorized logistic regression model was derived and its adequate prediction performance was confirmed (AUROC - 0.776; AUPR - 0.760).” The categorized model is NOT better than that in Table 3.

(2)    Again, the categorized model is neither superior to RF and Xgboost (Tree-based model) in Table 2. The strength of categorized model is not obviously seen.

(3)    Other points: abbreviation should be defined, e.g., BUN. Important predictors, MAP, body temperature, pulse pressur, were not shown in Table1. Please check carefully. 

Author Response

  1. Introduction:

(1)    For literature review, more evidence regarding machine learning methods can be provided by adding relevant studies, such as

Liu, L. P., et al. (2021). Machine learning for the prediction of red blood cell transfusion in patients during or after liver transplantation surgery. Frontiers in medicine, 8, 81.

McCluskey SA, et al. (2006). Derivation of a risk index for the prediction of massive blood transfusion in liver transplantation. Liver Transpl 12:1584–93. doi: 10.1002/lt.20868.

Pustavoitau A, et al. (2017) Predictive modeling of massive transfusion requirements during liver transplantation and its potential to reduce utilization of blood bank resources. Anesth Analg. 124:1644–52. doi: 10.1213/ANE.0000000000001994

 Besides, these references can be discuss in Discussion.

Thank you for the kind comments. We cited and discussed the above studies in the introduction as below:

Line 38-40 ; The incidence of major hemorrhage and transfusion during LT over the past decade has decreased significantly, but major bleeding during surgery is commonly expected [1].

Line 54-56 ; If the amount of bleeding can be predicted preoperatively, unnecessary transfusion preparation or catheterization can be reduced [2].

We also added relevant studies in the discussion as below:

Line 202-204 ; MELD was associated with patients requiring blood products [3], but was a poor predictor of blood loss or blood transfusion in previous studies.

[1] Liu, L.P.; Zhao, Q.Y.; Wu, J.; Luo, Y.W.; Dong, H.; Chen, Z.W.; Gui, R.; Wang, Y.J. Machine Learning for the Prediction of Red Blood Cell Transfusion in Patients During or After Liver Transplantation Surgery. Front. Med. 2021, 8, 1–9, doi:10.3389/fmed.2021.632210.

[2] Pustavoitau, A.; Lesley, M.; Ariyo, P.; Latif, A.; Villamayor, A.J.; Frank, S.M.; Rizkalla, N.; Merritt, W.; Cameron, A.; Dagher, N.; et al. Predictive Modeling of Massive Transfusion Requirements During Liver  Transplantation and Its Potential to Reduce Utilization of Blood Bank Resources. Anesth. Analg. 2017, 124, 1644–1652, doi:10.1213/ANE.0000000000001994.

[3] McCluskey, S.A.; Karkouti, K.; Wijeysundera, D.N.; Kakizawa, K.; Ghannam, M.; Hamdy, A.; Grant, D.; Levy, G. Derivation of a risk index for the prediction of massive blood transfusion in  liver transplantation. Liver Transplant. 2006, 12, 1584–1593, doi:10.1002/lt.20868.

  1. Methods:

Section 2.1:

(1)    Please give the total number of candidate predictors. Is it 59 shown in Figure 1? The author can provide a list of all predictors in Appendix.

Thank you for your kind comments. We added 59 features listed in order of their importance as Supplementary Table 1.

Section 2.2.2:

(1)    It appears that the feature selection method in favor of logistic regression model (e.g., These features were then further filtered using multivariable logistic regression.). With regard to the impact of feature selection might alter their findings, I suggest to include all significant predictors, 20 variables in Table 1, to re-run the prediction using these ML methods.  

In general, including all significant features results in poor machine learning models due to potential overfitting. Therefore, we conducted an unbiased feature selection. Per the reviewer’s request, we conducted machine learning training using all significant (17) features including the selected features in the original manuscript. The results (shown below) illustrate that having more features does not necessarily result in better models. Indeed, the multivariable logistic regression, which is the most parsimonious and interpretable, is still one of the best performing models, reassuring the robustness of our approach.

Machine learning method

Test dataset

AUROC

AUPR

Selected features

All significant features

Selected features

All significant features

Multivariable logistic regression

0.840

0.828

0.821

0.812

Elastic net

0.764

0.672

0.678

0.625

Random forests

0.803

0.825

0.783

0.808

Extreme gradient boosting

0.806

0.801

0.797

0.786

Neural networks

0.851

0.796

0.804

0.768

SVM with radial kernel

0.832

0.820

0.804

0.792

SVM with linear kernel

0.841

0.848

0.818

0.818

Section 2.2.3

(1)    The setting of parameters for each machine learning method (i.e., elastic net, RF..etc.) should be given.

The detailed model specifics are as below. We added this as a supplementary table in the revised manuscript.

Supplementary table 2. machine learning model specifics.

Method

Hyperparameter

Model specification and search grids

Selected value

Elastic net

: 500 equally spaced values in logarithmic scale between 10-6 and 1010

: 0, 0.1, 0.2, 0.3, 0.4, 0.5, 0.6, 0.7, 0.8, 0.9, 1

: 0.1813498

: 0

Random forests

mtry: 1, 2, 3, 4, 5, 6, 7, 8

mtry: 1

Extreme gradient boosting

Number of rounds: 1000 (early stopping with patience 20)

Maximum depth: 3, 5

Collection sample bytree: 0.7

eta: 0.05, 0.1

gamma: 0, 1, 2

Minimum sum of instance weight (hessian) needed in a child : 3, 5

subsample: 0.5

Iteration : 21

Maximum depth: 3

Collection sample bytree: 0.7

eta: 0.05

gamma: 2

Minimum sum of instance weight (hessian) needed in a child : 5

subsample: 0.5

Neural networks

Input layer with 8 nodes, two hidden layers with varying numbers of nodes with the relu activation, output layer of one node with the sigmoid activation, dropout with varying rates applied for both hidden layers, and epochs set to 1000 with early stopping patience 20.

Dropout: 0.2, 0.3

Batch size: 8, 16, 32,

Number of nodes in hidden layers: 10, 15, 20.

Dropout: 0.3

Batch size: 32

Number of nodes in hidden layers: 10

(2)    Typo: linear regression? or logistic regression

We changed linear regression to logistic regression. Thank you.

Section 2.2.4:

(1)    It is difficult to understand the scoring system. Please provide a diagram for this.

Thank you for your comments. The specific scoring system was shown in Figure 3. The legend of the Figure 3 was accidentally left out during the editing process, and so we revised it.

As a working example, consider a hypothetical patient with hepatocellular carcinoma, aPTT of 45 seconds, expected operation time of 9 hours, body temperature of 36.5℃, MELD score of 15, the last blood pressure in the ward before the surgery of 80/55/63 mmHg, and creatinine of 0.8 mg/dL. According to our scoring system, Table 4, aPTT of 45 seconds yields a score of +5, MELD of 15 a score of +4, MAP of 63mmHg a score of +14, and pulse pressure (the difference between systolic and diastolic blood pressure) of 35 a score of + 4. The total score thus sums to 27. According to Figure 3, the score 27 belongs to the score bin of 27-30, which predicts that the probability of bleeding more than 5000cc is 78.1%.

(2)    Categorizing continuous data looks pretty similar with the rationale of Tree-based methods, e.g., RF or Xgboost. The authors should compare the risk scoring system with that obtained from these tree-based methods.

Thank you for the sharp comments. In Table 2, we presented the performance comparisons among the various machine learning algorithms used. The tree-based methods, RF and Xgboost, showed AUROC of 0.803 and 0.806 and AUPR of 0.783 and 0.797, respectively. The final risk scoring system was associated with AUROC of 0.775 and AUPR of 0.753. As expected, the tree-based ensemble algorithms exhibited superior predictive performance compared to the scoring system. The drawback of these ensemble models, however, is in its low interpretability. The risk scoring system, despite its slightly inferior performance, is highly interpretable and can be easily used by the practicing physicians.

(3)    The authors can discuss the impact of small sample size (e.g., 30 % of validation data) on risk scoring.

Thank you for the comments. The risk scoring system was not developed using the test data (a 30% sample chosen randomly from the original data). Hence, it is unlikely that the size of the test data would have much impact on the performance of the risk scoring system. We believe that it is standard practice to randomly split the data into training and test datasets in a ratio of 7:3 or 8:2 before beginning the machine learning analysis.

  1. Results:

(1)    In Table 2. Logistic regression with AUROC=0.84 and AUPR=0.821. The final logistic model is presented in Table 3. It is not clear that “Firstly, an optimally categorized logistic regression model was derived and its adequate prediction performance was confirmed (AUROC - 0.776; AUPR - 0.760).” The categorized model is NOT better than that in Table 3.

Thank you for the comments. The purpose of developing the risk scoring system was not to improve the predictive performance. As the reviewer rightly said, the multivariable logistic performed better. In fact, most of the tested machine learning models (shown in Table 2) showed better performance. Instead, our aim in proposing a simpler (at the cost of a slightly lower predictive performance) risk scoring system was to “maximize the clinical utility”. Please note that the most widely used clinical scoring systems (such as Child-Pugh score, Apgar score, Apfel score, and etc.) are not necessarily the best performing models. Rather, the main advantage associated with these models is their easy applicability in a busy clinical setting. It would be impractical to ask the physician to substitute features expressed on a continuous numerical scale into a logistic regression model, calculate the output probability, and then categorize the most likely outcome.

(2)    Again, the categorized model is neither superior to RF and Xgboost (Tree-based model) in Table 2. The strength of categorized model is not obviously seen.

Thank you for the comments. We believe that we have sufficiently expressed our view on this issue in the response given to the immediately preceding comment. To repeat, we aimed at clinical utility rather than predictive performance.

The main strength of the categorized model is in its simplicity – the categorized variables have been mapped to integer-valued scores, which can be summed to yield the final score.

(3)    Other points: abbreviation should be defined, e.g., BUN. Important predictors, MAP, body temperature, pulse pressure, were not shown in Table1. Please check carefully.

Thank you for your kind comments. The authors added the full name of BUN. Also, mean arterial pressure, body temperature, and pulse pressure were added in Table 1, which are important predictors in the final logistic regression model. Operation time is 617 minutes ± 138.5 in EBL ≤ 5000 cc group and 666.2min ± 168.8 in EBL >5000 cc group. They are significantly different in the two groups. Mean arterial pressure before the surgery at the ward or intensive care unit was lower in EBL >5000 cc group (86.5 mmHg ± 10.7 vs 91.1 mmHg ± 10.6)

Table 1. patients’ demographics.

EBL≤5000cc

(N=228)

EBL>5000cc

(N=186)

p value

EBL (mean (SD))

2753.6 (1258.0)

8755.1 (4220.7)

<0.001

Age (mean (SD))

56.07 (8.5)

55.43 (9.1)

0.458

Sex = male (%)

159 (69.7)

132 (71.0)

0.869

Height (mean (SD))

165.20 (8.1)

165.27 (8.9)

0.932

Weight (mean (SD))

65.28 (11.0)

66.67 (11.8)

0.216

Emergency (%)

43 (18.9)

51 (27.4)

0.051

Cadaver donor (%)

41 (18.0)

44 (23.7)

0.194

Operation time (mean (SD))

617.5 (138.5)

666.2 (168.8)

<0.001

Liver cirrhosis (%)

141 (61.8)

148 (79.6)

<0.001

Alcoholic liver disease (%)

60 (26.3)

77 (41.4)

0.002

Hepatocellular carcinoma (%)

143 (62.7)

65 (34.9)

<0.001

MELD (mean (SD))

13.45 (8.5)

19.08 (10.7)

<0.001

Hb (mean (SD))

10.93 (2.2)

9.76 (2.0)

<0.001

Hct (mean (SD))

32.22 (6.3)

28.81 (5.8)

<0.001

Platelet (mean (SD))

94.60 (53.3)

80.98 (54.3)

0.011

PT (mean (SD))

1.4 (0.6)

1.7 (0.5)

<0.001

aPTT (mean (SD))

40.0 (13.2)

51.4 (30.0)

<0.001

Albumin (mean (SD))

3.1 (0.6)

2.9 (0.5)

<0.001

Blood Urea Nitrogen (mean (SD))

17.6 (11.0)

27.3 (23.0)

<0.001

Creatinine (mean (SD))

0.8 (0.4)

1.2 (1.1)

<0.001

eGFR (mean (SD))

87.0 (43.0)

56.2 (40.9)

<0.001

Mean arterial pressure (mean (SD))

91.1 (10.6)

86.5(10.7)

<0.001

Body temperature (mean (SD))

36.6 (0.4)

36.5 (0.4)

<0.001

Pulse pressure (mean (SD))

57.5 (12.5)

52.9 (12.6)

<0.001

Reviewer 2 Report

Thank you for this interesting topic. ML is certainly a future feature of perioperative care and intensive care medicine. Nevertheless I have some topics which still have to be discussed:

Introduction: How can you quantify "many"? espaecially for the statement that many patients experience mortality?

line 54-56: This is a keen statement. You cannot go from an overall risk of a certain population to the individual risk for each patient. even if the risk is low, if it meets the patient it hits him/her 100%. So it is not possible to rule out patients who do not need a central venous catheter or blood products.

Methods: line 77: "Variables known to be related to desease" Where from are they known?

Why did you look for CRRT but not for chronic kidney desease or hepatorenal syndrome?

Which MAP did you choose? was it a single measurement or an observation over time? Did you take a mean out of more measurements. What about patients on vasopressors? Please give an explanation for your rationale.

results: line 153/154: what is the difference between hepatic failure etiology and HCC status since HCC is a part of hepatic failure etiology?

which aptt was chosen and how did you deal with patients who received coagulation factors?

Which creatinine did you chose. which creatinine was used for patients on CRRT?

Table 1: Did you screen for fibrinogen levels? What was the result? if not so please explain.

Discussion: line 221ff: Degradation of vWF is reduced in chronic liver desease. So that there is a rebalancing of coagulation. Could you please comment on this in the discussion.

MAP and pulse pressure per se is difficult to use since there are many factors that have an influence on it (eg. fluid status, vascular resistance, inotropy, atherosclerosis, aortic valve insufficiency, vasopressor dose etc.). Of course in trauma patients, where you know, that the patient is bleeding this may be applicable but this is not applicable for preoperative patients. If you can explain in detail how MAP and pulse pressure f a non bleeding patient can be transfered to a prediction model please do so.

line 247ff: is there a proof for this statement?

line 252+253: the reference are not suitable for this. Please change!

Author Response

Introduction: How can you quantify "many"? espaecially for the statement that many patients experience mortality?

We highly appreciate the reviewer' insightful and helpful comments. Patients who underwent liver transplantation experience a 5-year mortality of 18.8% (ref : Kwong, A.J.; Ebel, N.H.; Kim, W.R.; Lake, J.R.; Smith, J.M.; Schladt, D.P.; Skeans, M.A.; Foutz, J.; Gauntt, K.; Cafarella, M.; et al. OPTN/SRTR 2020 Annual Data Report: Liver. Am. J. Transplant. 2022, 22 Suppl 2, 204–309, doi:10.1111/ajt.). But, because survival after liver transplantation can vary among health care centers, we agree that the expression of “many” patients can be ambiguous. We thus corrected the sentence in the introduction (line 36-38) as below:

Although liver transplantation (LT) has emerged as the treatment of choice for patients with end-stage liver disease, patients who underwent LT experience a variety of complications, including infection, mortality, and surgical re-intervention.

line 54-56: This is a keen statement. You cannot go from an overall risk of a certain population to the individual risk for each patient. even if the risk is low, if it meets the patient it hits him/her 100%. So it is not possible to rule out patients who do not need a central venous catheter or blood products.

Thank you for your critical comment. Indeed, once the event occurs, it becomes 100% sure on. However, before the event, the best we can do is to predict the probability of the event occurrence ahead of time to be prepared. Agreeing with your opinion, even though the calculated probability of bleeding more than 5000cc based on this paper is low, the anesthesiologist will prepare large-bore central venous catheter and blood products in the real practice. The predictive model proposed in our study would be more helpful for patients with a high predicted probability of massive bleeding since it would allow anesthesiologists and surgeons to be more prepared with actions and protocols under massive bleeding. The authors think it will be of great help to actual practice to find and predict risk factors for massive bleeding as one of evidence-based medicine, rather than predicting bleeding based on the current anesthesiologist's personal experience.

Methods: line 77: "Variables known to be related to disease" Where from are they known?

Thank you for your helpful comments. We added these features in Supplementary table 1. We corrected this sentence as below.

Methods (line 77-78 ) : The physiologic parameters induced by end-stage liver disease and outcomes after LT were collected via electrical medical records (EMR).

Why did you look for CRRT but not for chronic kidney desease or hepatorenal syndrome?

Thank you for your kind comments. It was reported that patients on CRRT during the intraoperative periods receive more blood transfusions (ref : Safwan M, Gosnell J, Collins K, Rizzari M, Yoshida A, Abouljoud M, Nagai S. Effects of Intraoperative Continuous Renal Replacement Therapy on Outcomes in Liver Transplantation. Transplant Proc. 2020, 52, 265-270. doi: 10.1016/j.transproceed). Therefore, in this study we sought to confirm the relationship between CRRT and massive bleeding although preoperative CRRT is not common (6.3% of 414 patients were on CRRT during the preoperative period).

Which MAP did you choose? was it a single measurement or an observation over time? Did you take a mean out of more measurements. What about patients on vasopressors? Please give an explanation for your rationale.

Thank you for your kind comments. For variables with multiple measurements, the values at the ward or intensive care unit closest to the surgery start time were assessed. A single measurement of the last blood pressure before departing to the operating room in the ward or intensive care unit was chosen (i.e. the last one of the routine check-ups of vital signs conducted three times a day in the ward). The same applies to vital signs such as the temperature and respiratory rate. Vasopressors were norepinephrine or vasopressin. For norepinephrine, the dose of norepinephrine was recorded by the flow rate (mcg/kg/min). We investigated the last flow rate if the patient had been receiving norepinephrine just before the surgery. For vasopressin, we investigated the flow rate(unit/hr) if the patient had been receiving vasopressin just before the surgery.

Added sentence (line 80-82): For variables with multiple measurements, the values at the ward or intensive care unit closest to the surgery start time were chosen. A single measurement of the last vital signs before departing to the operating room in the ward or intensive care unit was chosen.

results: line 153/154: what is the difference between hepatic failure etiology and HCC status since HCC is a part of hepatic failure etiology?

Thank you for your advice. Indeed, HCC is a part of hepatic failure etiologies. Other etiologies such as alcoholic liver cirrhosis, HBV or HCV carrier status were also included in feature selection. Feature selection revealed that HCC other than other etiologies is a predictive feature. Our results illustrate that that if patients without HCC will be given liver transplantation surgery, they have a higher risk of massive bleeding than those with HCC with the score increased by 5 points.

Discussion (Line 154-157) : After additional filtering through multivariable logistic regression analysis, eight features were finally selected : hepatocellular carcinoma (HCC) status, aPTT, operation time, body temperature, MELD, MAP, creatinine, and pulse pressure.

which aptt was chosen and how did you deal with patients who received coagulation factors?

Thank you for your helpful comments. We chose the last laboratory findings such as creatinine, aPTT and PT measured just before entering the operation room. For patients who received coagulation factors, the factors of transfused FFP or packed RBC in the preoperative period were investigated together.

Which creatinine did you choose. which creatinine was used for patients on CRRT?

Thank you for your helpful comments. We chose the last laboratory findings such as creatinine, aPTT and PT measured before entering the operation room. Routinely, for elective liver transplantation, early morning on the day of surgery, the blood sampling is scheduled. The results from those measurements were chosen in this study. In the emergent surgery, the last results measured in the ward or ICU were chosen. For the patients on CRRT, creatinine level was chosen just before the surgery and the feature of CRRT (if the patient is on CRRT or not just before the surgery) was investigated together.

As an additional response to the above comments, the choice of the last preoperative sample as our feature is primarily due to its intended use in our scoring system. We wish to direct the reviewer’s attention to the obvious fact that there is no point assessing the risk of bleeding during or towards the end of surgery. While intraoperative MAP, for example, would likely be more representative of surgical bleeding, it certainly cannot be used as part of a preoperative assessment. 

Table 1: Did you screen for fibrinogen levels? What was the result? if not so please explain.

Thank you for your kind comments. We initially obtained the data on fibrinogen levels. However, 165 of 414 patients did not have the data on preoperative fibrinogen. In Severance Hospital, the fibrinogen level of the preoperative period has been checked since 2017, and our data included patients who underwent LT between 2015 and 2020. Training a machine learning model using a feature with many missing values ​​is likely to decrease the reliability of the model. Furthermore, fibrinogen is often not part of the routine surveillance tests in many medical centers, and including such “expensive” information in a scoring system could hinder its widespread use. Therefore, we did not use the ‘fibrinogen’ feature.

Discussion: line 221: Degradation of vWF is reduced in chronic liver desease. So that there is a rebalancing of coagulation. Could you please comment on this in the discussion.

Thank you for your helpful comments. As you mentioned, in patients with liver cirrhosis, degradation of vWF is reduced and factor VIII is increased(ref : Lisman T, Porte RJ. Rebalanced hemostasis in patients with liver disease: evidence and clinical consequences. Blood. 2010,  12, 116, 878-85. doi: 10.1182/blood-2010-02-261891). APTT/PT may not be a reliable predictor to assess the status of coagulation because these laboratory tests are not sensitive for the deficiencies of the anticoagulants. The authors added that comments in the discussion as below :

Discussion(line 225 - 229) : Our results were consistent with those of previous reports. However, some papers have shown that coagulation defects of PT-INR or aPTT are inadequate to reflect the balance of coagulation due to the rebalancing of coagulation with increased von Willebrand factor and factor VIII and decreased natural anticoagulants proteins in patients with end-stage liver disease

MAP and pulse pressure per se is difficult to use since there are many factors that have an influence on it (eg. fluid status, vascular resistance, inotropy, atherosclerosis, aortic valve insufficiency, vasopressor dose etc.). Of course in trauma patients, where you know, that the patient is bleeding this may be applicable but this is not applicable for preoperative patients. If you can explain in detail how MAP and pulse pressure of a non bleeding patient can be transfered to a prediction model please do so.

Thank you for the comment. As you mentioned, MAP and pulse pressure would clearly change with conditions such as time, vasopressor use, or patients’ cardiac disease. Our view, however, is that whatever the factors that would have affected them, MAP and pulse pressure would still provide critical information about the current hemodynamic status of the patient (which is why they are referred to as ‘vital signs’). From a phenomenological point of view, we found that patients with EBL > 5000 cc group clearly exhibited a significantly lower MAP (86.5 ± 10.7 vs 91.1 ± 10.6) and pulse pressure (52.9 ± 12.6 vs 57.5 ± 12.5). Low MAP and pulse pressure can be observed in trauma patients and our massive bleeding patients, and they could have small circulating volume. Given that all measurements were collected in the preoperative period, there is no reason why this clear correlation should not be exploited for predictive purposes. To verify a causal relationship between MAP/PP, prospective RCT would be necessary.

As a last note, the patients with end-stage liver disease have a bleeding tendency due to the abnormalities in hemostasis. Therefore, physiologic responses of the cardiovascular system in these patients are likely similar to those with trauma with unidentified internal bleeding. Hence, we believe that lower MAP might still signify occult bleeding and the need for intraoperative RBC transfusion.

line 247ff: is there a proof for this statement?

Thank you for your careful comments. You recommended a proof for the statement that previous episodes of variceal bleeding and previous abdominal surgery were not valuable in predicting blood loss. Aziz at el. found that previous abdominal surgery history as a whole does not appear to impact the need for RBC and FFP transfusion in liver transplantation but recent abdominal surgery and upper abdominal surgery predict the need for greater blood product transfusion during liver transplantation (ref : Aziz A, Ito T, Younan S, Dinorah J 3rd, Agopian VG, Farmer DG, Busuttil RW, Kaldas FM. The Impact of Previous Abdominal Surgery in a High-Acuity Liver Transplant Population. J Surg Res. 2021, 258, 405-413. doi: 10.1016/j.jss.2020.08.064). In this study, previous abdominal surgery also has no significant association with massive bleeding.

line 252+253: the reference are not suitable for this. Please change!

Thank you for your careful comments. The authors correct the numbering for the references as below.

An increased bleeding tendency has been associated with various degrees of renal insufficiency. Renal insufficiency results from multifactorial causes but affects all aspects of platelet function, including adhesion, secretion, and aggregation [1]. According to Modanlou et al. [2], transplantation is almost four times more likely to require 10 units of RBC in cases where the recipient patient has a pre-surgery serum creatinine level greater than 1.3 mg/dL.

[1] Sohal, A.S.; Gangji, A.S.; Crowther, M.A.; Treleaven, D. Uremic bleeding: Pathophysiology and clinical risk factors. Thromb. Res. 2006, 118, 417–422, doi:10.1016/j.thromres.2005.03.032.

[2] Modanlou, K.A.; Grossman, B.J.; Oliver, D.A. Liver donor’s age and recipient’s serum creatinine predict blood component use during liver transplantation. Transfusion 2009, 49, 2645–2651, doi:10.1111/j.1537-2995.2009.02325.x.

Round 2

Reviewer 1 Report

I have no other comments.